# Resorbable Membrane Pouch Technique for Single-Implant Placement in the Esthetic Zone: A Preliminary Technical Case Report

**DOI:** 10.3390/bioengineering9110649

**Published:** 2022-11-04

**Authors:** Akiyoshi Funato, Chihiro Ishikura, Koji Naito, Akira Hasuike

**Affiliations:** 1Nagisa Dental Clinic, Kanazawa 920-0031, Japan; 2DIABUILDING Dental Clinic, Tokyo 104-0033, Japan; 3Department of Periodontology, Nihon University School of Dentistry, Tokyo 101-8310, Japan; 4Dental Research Center, Nihon University School of Dentistry, Tokyo 101-8310, Japan

**Keywords:** dental implant, resorbable membrane pouch techniques, case report, connective-tissue graft

## Abstract

The conventional protocol for lateral guided bone regeneration (GBR) in esthetic areas requires the securing of resorbable collagen membranes using titanium cortical bone pins to immobilize bone grafts. These procedures are highly invasive and can increase patient morbidity and discomfort. Herein, we introduce a minimally invasive novel resorbable membrane pouch technique, wherein collagen membranes can be immobilized by securing them to the periosteum without the need of titanium pins. We describe 11 cases of single-immediate- or delayed-implant placement in the atrophic maxilla esthetic zone. All implants were successful and functional without pain or inflammation and with optimal soft-tissue health and esthetics. Radiographic evaluation with cone-beam computed tomography (CBCT) and esthetic assessment using the pink esthetic score (PES) were performed. At the time of implant placement, the average augmented bone width was 2.8 ± 0.6 mm on CBCT analysis. In all cases, resorption of the augmented bone was confirmed with an average of −1.3 ± 0.8 mm. Soft-tissue outcomes were scored 1 year after permanent restoration. The PES score 1 year after treatment was 11.9 ± 1.4. The resorbable membrane pouch technique with immediate or delayed implant placement for buccal dehiscence in the esthetic area can be predictable and is minimally invasive.

## 1. Introduction

To achieve satisfactory esthetic results and longevity, several surgical options have been applied, especially in the anterior region of the maxilla, such as immediate implant placement [1] and partial extraction therapy [2]. These techniques are useful only in limited cases. The presence of facial lamellar bone is considered a prerequisite for both a high implant persistence rate and good esthetic results [3]. Buccal bone thickness should be at least 2 mm and ideally 4 mm to ensure adequate support of the soft tissue, to prevent resorption of the buccal bone wall following restoration, and to achieve optimal esthetic results [4]. However, the facial bone wall in the anterior maxilla is frequently thin or nonexistent, especially in cases where teeth are lost because of luxation injuries or periodontal or endodontic lesions. In cases of severe horizontal bone loss, horizontal augmentation of the maxillary alveolar ridge may be necessary and can be achieved using a variety of surgical approaches [5]. Guided bone regeneration (GBR) is a surgical technique that increases bone volume in areas planned for implant placement or around previously placed implants. The GBR procedure is based on the principles of guided tissue regeneration (GTR) in periodontal regeneration, which emphasizes the need for epithelial and fibrous cell exclusion to enable favorable bone regeneration [6]. In the GBR method, mechanical barriers are used to create a space to be filled with new bone [7]. Mechanical barriers must satisfy certain physicochemical characteristics to ensure biocompatibility, tissue integration, cell adhesion, and space-making ability, as well as ease of clinical use. 

Titanium mesh was used for GBR even before dental implant treatment became widespread [8]. The chief purpose of this technique is to confine graft materials to the recipient site to reconstruct alveolar ridges. Titanium has excellent mechanical properties for the stabilization of bone grafts, high strength and rigidity, and resistance to corrosion. However, since titanium mesh is microporous and far from occlusive, there is a possibility that fibrous tissue could intrude into the bone defect area and jeopardize bone regeneration [9]. Furthermore, oral biofilm formation on the titanium materials has been widely reported [10].

Membranes, which can be resorbable or nonresorbable, have also been used as mechanical barriers in GBR procedures. Polytetrafluoroethylene (PTFE) is the most well-known barrier membrane for both GTR and GBR. One drawback in the use of this type of membrane is the necessity for its removal with a second-stage surgical procedure [11].

Resorbable collagen membranes exhibit better soft-tissue compatibility than nonresorbable membranes. However, a drawback of resorbable barrier membranes is that maintaining the shape of augmented bone grafts without collapse is difficult. Thus, resorbable collagen membranes should be pulled back over the mixture of bone grafts, stretched, and held in place using titanium cortical bone pins to immobilize the grafts [12]. In this surgical protocol, titanium pins must be removed during secondary surgery [13]. These complex procedures can be highly invasive and can increase patient morbidity and discomfort [14]. 

Therefore, the present study aimed to introduce a novel and minimally invasive GBR technique. In this technique, collagen membranes are immobilized by being secured to the periosteum without placing titanium pins. In this preliminary report, 11 maxillary anterior cases were examined using cone-beam computed tomography (CBCT) analysis and esthetic scoring.

## 2. Case Presentation

### 2.1. Patient Selection

This study was designed as a retrospective case series evaluation without any controls, aimed to provide a foundation for further studies with larger sample sizes and a longer follow-up duration. This report was conducted in accordance with the Declaration of Helsinki. Owing to the retrospective nature of the case series report, ethical committee approval was not needed. Written informed consent was obtained from all patients for publication of this case series report. 

Patients had to meet the following inclusion criteria to undergo this therapy: need for single-implant placement in the esthetic zone of atrophic maxilla (central or lateral incisors), aged 25 years or older, partially or completely missing buccal bone, acceptable oral hygiene, feasible implant placement inside the bone housing with preserved bone height at the palatal site, and agreement to undergo CBCT before and after surgery and after the delivery of the superstructure. The exclusion criteria were uncontrolled systemic conditions that may affect the outcome of surgical treatment, such as diabetes, bone metabolic disease, or pregnancy. Smokers were not eligible.

The surgeon selected eleven patients (eight women and three men, with a mean age of 45.5 ± 9.5 years) who visited a private practice in Japan for dental implant treatment between December 2013 and August 2016. 

### 2.2. Treatment Procedures

All surgical procedures were conducted by the same surgeon (A.F.) under local anesthesia with a solution of 2% xylocaine and 1:80,000 epinephrine. In cases of severe acute infection (edema, suppuration, abscess, and spontaneous bleeding), delayed implant placement was selected. When disharmony of the gingival margin of the tooth to be replaced with that of adjacent teeth was confirmed, delayed implant placement was performed to obtain a sufficiently wide keratinized mucosa. Otherwise, immediate implant placement was performed.

A schematic diagram of the resorbable membrane pouch technique is shown in Figure 1. In cases of immediate placement, the teeth were extracted as the first surgical step before making any incision. An intracrevicular incision was selected as an internal bevel incision for the natural teeth. To permit broad flap reflection, a triangular flap design was used for all cases, with one vertical releasing incision on the distal side of the canine in the same quadrant. The intracrevicular incision was extended to the vertical releasing incision on one side and one tooth distal on the opposite side. In cases of delayed implant placement, a crestal incision was made connecting the angles of the intracerebroventricular incision in both adjacent teeth. At the implant placement site, a full-thickness flap was elevated, and the bone crest was accurately exposed for only 3–4 mm (Figure 1a). A full-thickness flap was connected to a partial-thickness flap in the basal part of the maxilla (Figure 1a). Great care was taken not to perforate the flaps in the connection between the full-thickness flap and partial-thickness flap. The periosteum attached to the buccal basal part of the maxillary bone at the implant placement site was segregated using a bone chisel to prepare periosteum tabs for membrane fixation (Figure 1a). Periosteum tabs were prepared just apical to the buccal dehiscence in lengths of 2–3 mm. Following the preparation of a pouch for lateral GBR, prosthetically driven crestal implant (T3 Implants, Biomet 3i, Palm Beach Gardens, USA or OneQ, Dentis, Daegu, Korea) placement was performed.

A resorbable membrane (OsseoGuard^®^, Biomet 3i, Palm Beach Gardens, FL, USA or Ossix^®^, ColBar R & D Ltd., Ramat Hasharon, Israel) was trimmed to fit and was inserted beneath the periosteum tab (Figure 1b). The lateral edges of the membrane were seated beneath the periosteum of adjacent teeth. The membrane was secured to the periosteum using resorbable sutures (Vicryl 6.0 suture, Ethicon, Somerville, MA, USA) (Figure 1b). The internal space of the pouch surrounded by the resorbable membrane and exposed labial implant surface was treated with demineralized bovine bone mineral (DBBM) (Bio-Oss, Geistlich, Wolhusen, Switzerland), which was covered with a resorbable membrane. If the membrane could not fully overlap, an additional membrane was applied. The membrane was secured to the palatal wall using resorbable sutures (Vicryl 6.0 suture, Ethicon, Somerville, MA, USA; Figure 1b). 

To achieve good esthetic soft-tissue results, a connective-tissue graft was also placed during the same surgery (Figure 1c). By preserving the periosteum, connective-tissue grafts were secured only during a single surgical intervention. A connective-tissue graft obtained from a palatal site was placed on the buccal superior aspect and secured to the periosteum or mucosal flap. In cases of immediate placement, transmucosal abutments were mounted.

Finally, a mucoperiosteal flap in the delayed-implant-placement case was secured using a horizontal mattress suture and a single interrupted suture. Mucosal flaps in both delayed- and immediate-implant-placement cases were secured using only single interrupted sutures. The vertical incision was closed with a single interrupted suture. Resorbable materials (Monocryl 5-0, Ethicon or Vicryl 6.0 suture, Ethicon, Somerville, MA, USA) were used for all sutures, which were removed after 2–3 weeks. The patients were instructed to avoid brushing in the surgical area for at least 14 days postoperatively. All patients received a prescription for a chlorhexidine (0.12%) rinse. Postoperative antibiotics (amoxicillin 500 mg or clindamycin 300 mg) and anti-inflammatory drugs (ibuprofen 600 mg) were prescribed. The remaining procedures were performed according to standard clinical practice.

Eleven patients (eight women and three men, with a mean age of 45.5 ± 9.5 years) received single dental implant placements. The details of all cases are provided in Table 1. No implants exhibited any adverse or unanticipated events. Preoperative and perioperative photographs of representative immediate case #2 are presented in Figure 2. The average duration of nonloading was 19.7 ± 4.1 weeks.

### 2.3. Radiographic Assessment

To assess the alterations in the alveolar ridge, CBCT images of the implant sites were acquired immediately after surgery and after permanent restoration using Trophy Pan Pro (Yoshida, Tokyo, Japan). The clinical scanning protocol was fixed to a 10 × 10 cm field of view, a voxel size of 90 μ, 360° rotation, resulting in 17.5 s scanning time, at 90 kVp and 5 mA. A cross-sectional slice according to the long axis of the implant was generated by using actual implant lengths as a reference. Two reference lines were subsequently drawn. The horizontal bone thickness from the implant surface at the first thread to the outermost edge of the buccal bone was measured (Figure 3). The radiological measurements were performed by an independent examiner (C.I.)

At the implant placement, the average augmented bone width was 2.8 ± 0.6 mm in CBCT analysis. In all cases, an average augmented bone resorption of −1.3 ± 0.8 mm was confirmed. CBCT images of immediate case #2 at implant placement and permanent restoration are shown in Figure 4a,b, respectively.

### 2.4. Esthetic Evaluation

Soft-tissue outcomes were scored 1 year after permanent restoration. The pink esthetic score (PES) proposed by Furhauser et al. [13] was chosen as the criterion for determining the soft-tissue esthetic outcome of the implant site. PES includes seven variables: mesial papilla, distal papilla, soft-tissue level, soft-tissue contour, alveolar process deficiency, soft-tissue color, and texture. Using a 0–1–2 scoring system, where 0 is the lowest and 2 is the highest, the maximum achievable PES is 14. The threshold for an acceptable PES was 8. Scores ≥ 12 indicated a nearly perfect outcome. All PES evaluations were completed by a clinician who had not participated in any related therapy. The average PES score at 1 year after treatment was 11.9 ± 1.4. A clinical view of immediate case #2 at permanent restoration is shown in Figure 4c,d. Clinical photographs and CBCT images of representative delayed-implant-placement cases are displayed in Figure 5.

## 3. Discussion

Lateral GBR procedures aim to reconstruct deficient alveolar ridges or build peri-implant dehiscence. One systematic review reported that an intervention combining bone grafts with barrier membranes was associated with superior outcomes [5]. Although autografts are considered the gold standard among bone-graft substitutes, they have the disadvantages of greater invasiveness during the procedure and a limited quantity of grafts that can be harvested. DBBM is the most frequently used nonautogenous graft because of its osteoconductive property [15]. Our novel technique used DBBM as a bone graft in conjunction with resorbable membranes. Thus, we selected DBBM as a bone-graft substitute for minimal-intervention lateral GBR in this novel technique.

The most impressive point of the present technique is that we created a partial-thickness flap and used periosteum tabs to secure the resorbable membrane. In the conventional GBR technique, full-thickness flap elevation with a periosteal-releasing incision is commonly used to elevate a tensionless flap. Nevertheless, this flap design often results in complications, such as swelling, bleeding, patient discomfort, flap perforation, and graft exfoliation [16,17]. One of the main causes of these complications is the placement of deep periosteal incisions, which interrupt periosteal blood-vessel circulation. Increased tissue swelling due to postoperative blood stasis generates tension at the crestal incision line, which, in turn, may compromise wound healing and lead to premature membrane exposure [18]. In our proposed surgical technique, the preparation of a partial-thickness flap reduced the soft-tissue tension. Furthermore, in the conventional GBR protocol, a titanium bone pin is used for the fixation of barrier membranes. Bone pins pose a risk of perforating anatomical structures, such as the inferior alveolar nerve [13]. The removal of pins or screws also poses risks, such as bone loss, scar formation, and surgical complications [19]. Urban et al. introduced a periosteal suturing technique with resorbable sutures for fixation of grafts and membranes without using titanium pins [20]. The critical difference between their technique and the present technique is the flap design. They reflected a full-thickness flap and used the internal periosteum as the anchor for vertical mattress sutures. In their technique, the blood supply around the adjacent teeth could be interrupted by creating a full-thickness flap. Since sufficient blood supply is a critical factor for successful GBR [21], their technique would limit osteoprotective capacity through sufficient blood supply. Therefore, we used a partial-thickness flap for the present technique. 

The periosteum is a well-document potential source of osteogenic cells, growth factors, and blood. Our group examined the effects of the pedicle periosteum on bone regeneration in a rabbit calvarial-bone-defect model. Histological sections showed that the periosteum contributes to the formation of new bone by acting as a mechanical barrier and source of osteogenic factors [22]. From this perspective, our present approach using the periosteum tab would optimally utilize the properties of the periosteum. Steigman et al. also introduced a periosteal pocket-flap design for lateral GBR in the posterior area [23]. The critical difference between the present technique and theirs is the combined use of resorbable membranes. Because they only used a periosteal pocket flap without resorbable membranes, it was impossible to cover large defect sites. Furthermore, their procedure could not be universally performed in all anterior esthetic cases and may provide unpredictable outcomes in such cases.

In the CBCT analysis, augmented bone was confirmed in every case. However, the volume of augmented bone greatly decreased (−1.3 ± 0.8 mm). One possible reason for bone resorption is the morphology of bone dehiscence. Le et al. assessed the relationship between the vertical buccal dehiscence size and the outcome of implant placement in lateral GBR [24]. They showed that large dehiscence resulted in only partial improvement. Thus, future studies should assess the correlation between the dehiscence size and clinical results. Another potential reason for bone resorption is how well the space can be maintained under tissue pressure. We used connective-tissue grafts in all cases. Pressure propagated from the thick connective tissue may render the secured space fragile. The application of an appropriate connective graft and placing pressure on the resorbable membrane must also be important factors. 

On esthetic assessment, the PES score was very high in all cases (11.9 ± 1.4 points). Although only a single surgical intervention was performed for implant placement in defects with dehiscence, superior esthetic results were obtained in all cases. This is an advantage of the current surgical technique. A systematic review showed that buccal gingival thickness could be increased after a combination therapy of soft-tissue graft and immediate implant placement [25]. Simultaneous connective-tissue grafts seem to work well and may contribute to a high PES score. Comparative studies with a large sample size need to be conducted in the near future to assess the effect of soft-tissue grafts.

This study has some limitations. First, this was a retrospective case series that only studied the application effects of the resorbable membrane pouch technique for single-implant placement in a single center. Prospective research with control groups should be studied. Secondly, because of the nature of the case series, heterogeneity was confirmed in treatments among the 11 patients. Especially, the duration between the initial radiographic imaging and follow-up imaging varied from patient to patient. Thirdly, in the present report, histological assessments were not conducted. Histology of bone and soft tissues would reveal biological backgrounds of the present treatment.

Despite these limitations, the resorbable membrane pouch technique with immediate or delayed implant placement in the esthetic area could be a reliable, minimally invasive treatment option. In conventional procedures, cases of severe bone resorption in the buccal wall should be treated with delayed implant placement. However, it is possible to apply immediate implant placement to these compromised cases using the present surgical technique. As a routine surgical technique, surgeons should be cautious not to perforate the partial-thickness flap. If perforations are confirmed in the implant-supporting soft tissue, it is recommended to place a connective-tissue graft just beneath the perforation.

## 4. Conclusions

This case series demonstrated that the resorbable membrane pouch technique with immediate or delayed implant placement for buccal dehiscence in the esthetic area is a predictable, minimally invasive treatment option. Although all implants were successful without complications and with optimum esthetic results, this study has some limitations, including a small sample size with limited variables and a relatively short follow-up period. Therefore, to recommend this technique as a routine treatment, well-designed, randomized prospective clinical studies with longer observation periods must be conducted.

## Figures and Tables

**Figure 1 bioengineering-09-00649-f001:**
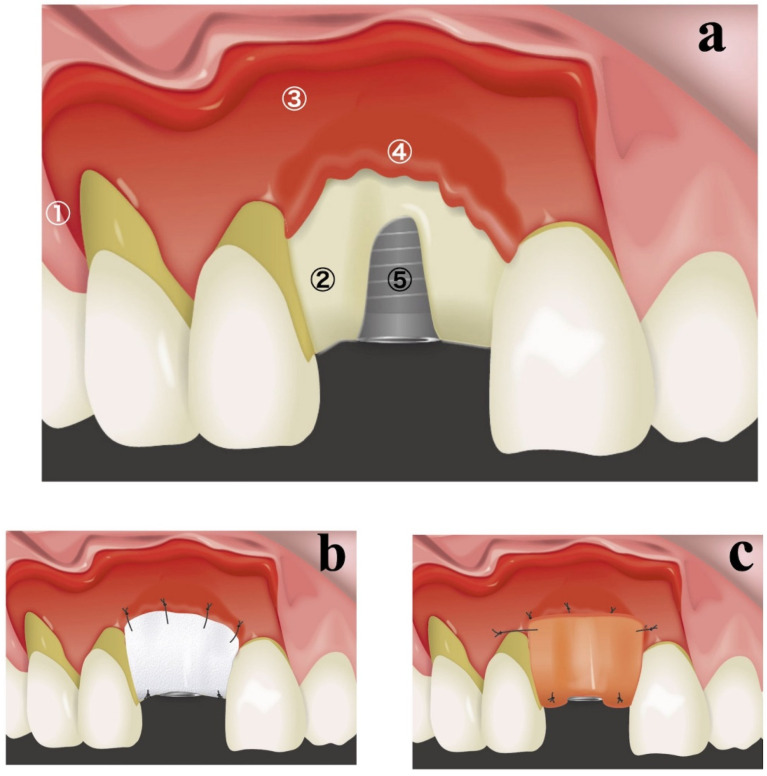
Illustrations of the surgical procedures. (**a**) ① A vertical releasing incision was made on the distal side of the canine in the same quadrant to create a triangular flap. ② At the implant placement site, a full-thickness flap was elevated and the bone crest was exposed. ③ A full-thickness flap was connected to a partial-thickness flap in the basal part of the maxilla. ④ Periosteum tabs were prepared just apical to the buccal dehiscence in lengths of 2–3 mm. ⑤ Following the pouch preparation, prosthetically driven crestal implant placement was performed. (**b**) A resorbable membrane was inserted beneath the periosteum tab. The membrane was secured to the periosteum using resorbable sutures. The internal space of the pouch, surrounded by the resorbable membrane and exposed labial implant surface, was treated with a bone graft. The membrane was secured to the palatal wall using resorbable sutures. (**c**) A connective-tissue graft obtained from the palatal site was placed on the buccal superior aspect and secured to the periosteum or mucosal flap.

**Figure 2 bioengineering-09-00649-f002:**
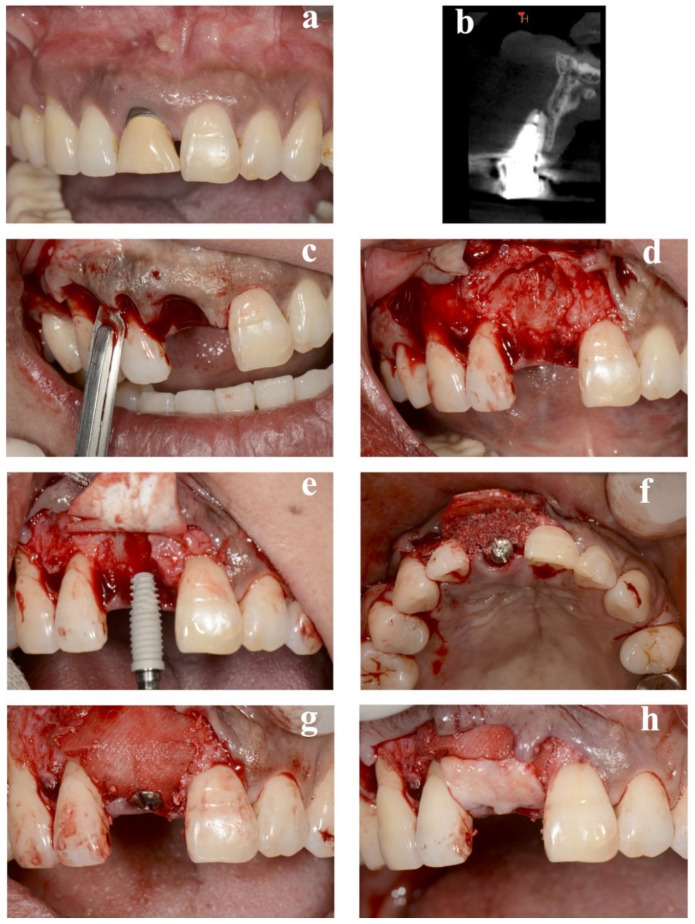
Replacement of a maxillary central incisor with an immediate implant in case #2. (**a**) Presurgical frontal view. (**b**) Cross-sectional view of preoperative CBCT. (**c**) Representation of the intracrevicular partial-thickness incision made around the natural teeth. (**d**) Preparation of periosteum tabs just apical to the buccal dehiscence in the lengths of 2–3 mm. (**e**) Insertion of a resorbable membrane beneath the periosteum tab. Execution of implant placement following the preparation of a pouch for lateral GBR. (**f**) Treatment of the internal space of the pouch surrounded by the resorbable membrane and exposed labial implant surface with a demineralized bovine bone mineral. (**g**) Covering of the surgical site with the membrane. (**h**) Securing of the connective-tissue graft to the periosteum. CBCT: cone-beam computed tomography, GBR: guided bone regeneration.

**Figure 3 bioengineering-09-00649-f003:**
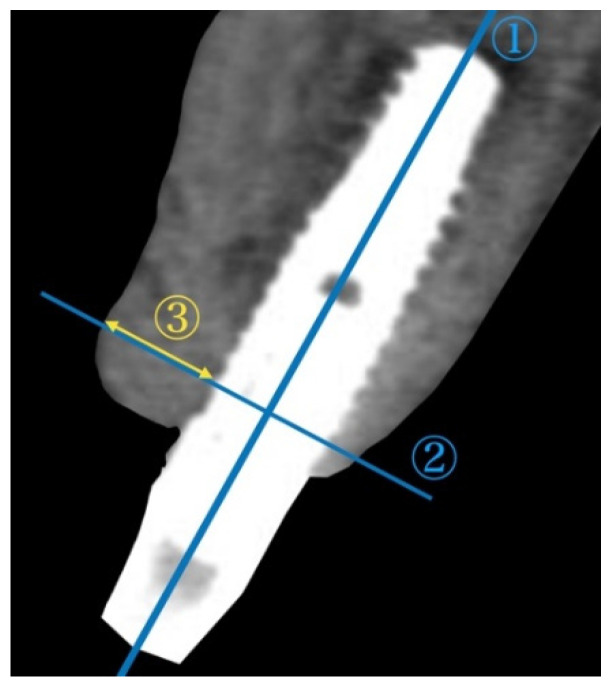
Assessment of the horizontal bone thickness with cross-sectional CBCT images. ① A vertical line was drawn in the center of the implant. ② The horizontal reference line was drawn perpendicular to the vertical line crossing the first thread of implant. ③ The horizontal bone thickness from the implant surface at the first thread to the outermost edge of the buccal bone was measured. CBCT: cone-beam computed tomography.

**Figure 4 bioengineering-09-00649-f004:**
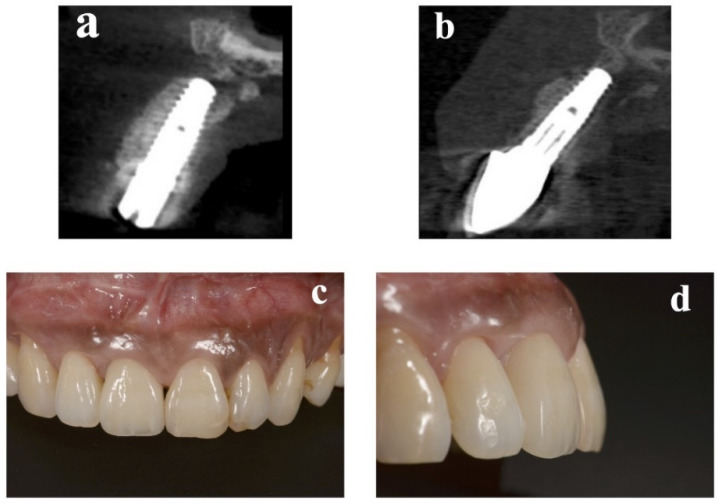
Replacement of a maxillary central incisor with an immediate implant in case #2. (**a**) CBCT at implant placement. (**b**) CBCT 1 year later. (**c**) Frontal view of the final implant restoration. (**d**) Buccal view of the final implant restoration. CBCT: cone-beam computed tomography.

**Figure 5 bioengineering-09-00649-f005:**
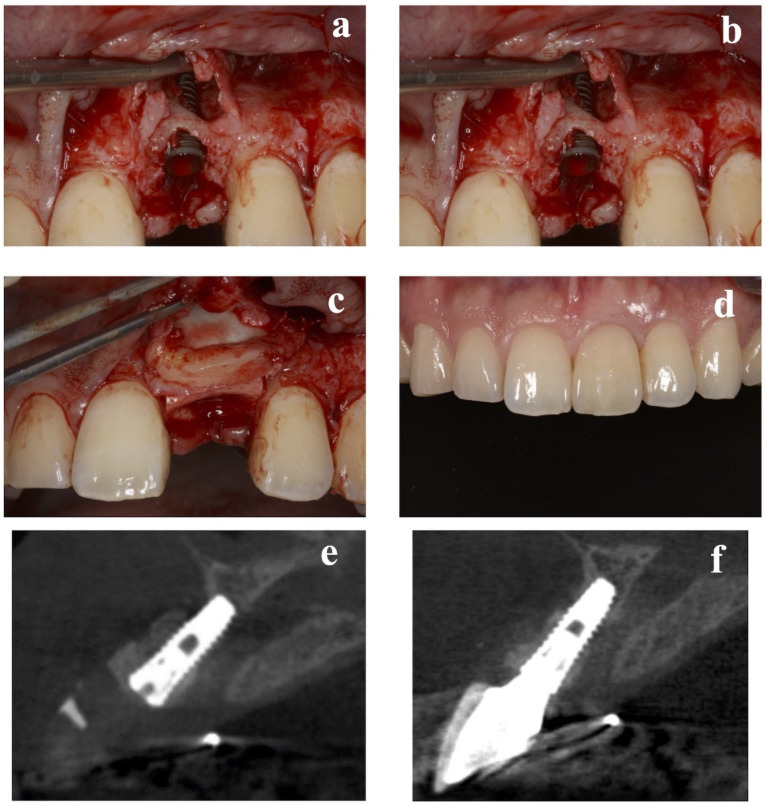
Replacement of a maxillary central incisor with a delayed implant in case #6. (**a**) Presurgical frontal view. (**b**) Placement of the implant 24 weeks after tooth extraction. (**c**) Securing the resorbable membrane and connective-tissue graft to the periosteum. (**d**) Frontal view of the final implant restoration. (**e**) CBCT at implant placement. (**f**) CBCT 1 year later. CBCT: cone-beam computed tomography.

**Table 1 bioengineering-09-00649-t001:** Case description and radiographic and esthetic results.

Case	Sex, Age (Years)	Tooth	Placement	Implant	Duration of Nonloading (Weeks)	Duration of Treatment (Months)	Bone Thickness in CBCT Analysis (mm)		PES after
							Baseline	Final	1 Year
1	Female, 59	11	Immediate	3i	20	26	2	1.2	13
2	Female, 55	11	Immediate	3i	16	9	3.4	1.3	13
3	Female, 48	11	Immediate	OneQ	20	11	3.2	2.6	14
4	Male, 38	21	Delayed (15 W)	3i	25	36	3.5	3.3	9
5	Female, 54	11	Delayed (12 W)	3i	17	21	1.6	1.4	12
6	Male, 27	21	Delayed (24 W)	OneQ	12	11	2.6	1.8	13
7	Male, 38	21	Delayed (28 W)	OneQ	27	18	3.1	2.3	11
8	Female, 49	21	Delayed (84 W)	OneQ	24	17	3	2.1	11
9	Female, 50	22	Delayed (19 W)	3i	24	11	3.2	1.3	11
10	Female, 46	22	Delayed (4 W)	3i	21	10	2	0.6	11
11	Female, 38	21	Delayed (16 W)	OneQ	18	8	3.2	1.7	13

CBCT: cone-beam computed tomography, PES: pink esthetic score, W: week.

## Data Availability

Not applicable.

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
