# Peer review of "Resorbable Membrane Pouch Technique for Single-Implant Placement in the Esthetic Zone: A Preliminary Technical Case Report"

_bioengineering, 2022, doi:10.3390/bioengineering9110649_

Round 1
Reviewer 1 Report
The purpose of this study was to propose a method of minimally invasive resorbable membrane pouch.
The findings may help the practitioner and the topic is within the scope of the journal.
Nevertheless, there are important things to highlight regarding this research:
Material & Methods
No specific details regarding patient consent. Your manuscript does not contain a complete IRB statement regarding ethics board approval. Original articles need to contain a statement about the Helsinki Declaration of 1975, as in the example given here: “This study was approved by the human subjects ethics board of XXXXX and was conducted in accordance with the Helsinki Declaration of 1975, as revised in 2013”.
Who selected the sample? Years of practice? Where were the radiographic analyses performed? There must be much more detail about how the images were measured. Please, include figures that show the measurements for those of us who have not done this by ourselves.
Discussion:
There is no Limitations section.
Author Response
Thank you very much for providing important comments. We are thankful for the time and energy you expended. Our responses to the referees’ comments are as follow:
#1. No specific details regarding patient consent. Your manuscript does not contain a complete IRB statement regarding ethics board approval. Original articles need to contain a statement about the Helsinki Declaration of 1975, as in the example given here: “This study was approved by the human subjects ethics board of XXXXX and was conducted in accordance with the Helsinki Declaration of 1975, as revised in 2013”.
→ Thank you for your comments. Our manuscript is the retrospective case series report. Thus, we do not have ethic committee or IRB approval. We regard it is not applicable to our manuscript. Written informed consent was obtained from all patients for publication of this case series report. I added line 80-85 & 321-323.
#2. Who selected the sample? Years of practice?
→ Thank you for the comments. I revised the description. Details of patient selection were written in line 86-96.
#3. Where were the radiographic analyses performed? There must be much more detail about how the images were measured. Please, include figures that show the measurements for those of us who have not done this by ourselves.
→ Thank you for the instruction. I added figure3, explaining the assessment. Description was also revised, line 181-189.
#4.There is no Limitations section.
→ Thank you for the comments. I added description regarding limitation of the present work, line 289-296.
Reviewer 2 Report
01
There are some sentences in the text without reference to a previous study (or studies) in order to give evidence to their statements. Without references, these statements would be mere assumptions or allegations by the authors of the manuscript. Therefore, each of the following sentences need at least one reference to back up their statement:
“In this surgical protocol, titanium pins must be removed during a secondary surgery.”
“These complex procedures can be highly invasive and can increase patient morbidity and discomfort.”
“One of the main causes of these complications is the placement of deep periosteal incisions, which interrupt periosteal blood-vessel circulation. Increased tissue swelling due to postoperative blood stasis generates tension at the crestal incision line, which, in turn, may compromise wound healing and lead to premature membrane exposure.”
“Bone pins pose a risk of perforating anatomical structures, such as the inferior alveolar nerve. The removal of pins or screws also poses risks such as bone loss, nerve lesions, scar formation, and surgical complications.”
“Since sufficient blood supply is a critical factor for successful GBR, (…)”
02
No approval from a local ethical committee was obtained, which is worrying.
03
Were the patients oriented to mouth rinse with chlorhexidine in the post-operative period, due to the lack of brushing in the surgical area for at least 14 days postoperatively?
04
Where is the reference number to the following statement?:
“Le et al. assessed the relationship between the vertical buccal dehiscence size and the outcome of implant placement in lateral GBR.”
05
“This is a chief advantage of the current surgical technique. It is considered that connective tissue grafts work very well in preserving soft-tissue volume and morphology.”
It is not possible to state that if the present case series did not have a control group.
Author Response
Thank you very much for providing important comments. We are thankful for the time and energy you expended. Our responses to the referees’ comments are as follow:
#1. There are some sentences in the text without reference to a previous study (or studies) in order to give evidence to their statements. Without references, these statements would be mere assumptions or allegations by the authors of the manuscript.
→ Thank you for your instruction. I added references.
#2. No approval from a local ethical committee was obtained, which is worrying.
→ Thank you for your comments. Our manuscript is the retrospective case series report.
Thus, we do not have ethic committee or IRB approval. We regard it is not applicable to our manuscript. Written informed consent was obtained from all patients for publication of this case series report. I added line 80-85 & 321-323.
#3. Were the patients oriented to mouth rinse with chlorhexidine in the post-operative period, due to the lack of brushing in the surgical area for at least 14 days postoperatively?
→ Thank you for your comments. All patients received a prescription for a chlorhexidine (0.12%) rinse. Postoperative an-tibiotics (amoxicillin 500 mg or clindamycin 300 mg) and anti‐inflammatory drugs (ibuprofen 600 mg) were prescribed. I added line142-146.
#4. Where is the reference number to the following statement?:
“Le et al. assessed the relationship between the vertical buccal dehiscence size and the outcome of implant placement in lateral GBR.”
→ I am very sorry to delete the reference. I added it.
#5. “This is a chief advantage of the current surgical technique. It is considered that connective tissue grafts work very well in preserving soft-tissue volume and morphology.”
It is not possible to state that if the present case series did not have a control group.
→ Thank you for your comments. I revised the expression. Please check line 281-288.
Reviewer 3 Report
The article is interesting, but some things need to be changed:
1) expand the introduction
2) add sentences about titanium implant infections at the beginning of the introduction part, Koopaie, Maryam, et al. "Advanced surface treatment techniques counteract biofilm-associated infections on dental implants." Materials Research Express 7.1 (2020): 015417.
3) specify the inclusion criteria in the materials and methods
4) more patients need to be included to get clearer data on this issue and to gain statistical significance
5) finally, you make a consideration based on histological analysis. Can you state on what basis you reached this conclusion? Was the biopsy performed and evaluated histologically? If so, you should post a picture and comment on it.
6) the English language needs to be improved.
Author Response
Thank you very much for providing important comments. We are thankful for the time and energy you expended. Our responses to the referees’ comments are as follow:
#1. expand the introduction
#2. add sentences about titanium implant infections at the beginning of the introduction part, Koopaie, Maryam, et al. "Advanced surface treatment techniques counteract biofilm-associated infections on dental implants." Materials Research Express 7.1 (2020): 015417.
→Thank you for your comments. I expanded introduction, introducing the above mentioned artcle.
3) specify the inclusion criteria in the materials and methods
→ Thank you for the comments. I revised the description. Details of patient selection were written in line 86-96.
4) more patients need to be included to get clearer data on this issue and to gain statistical significance
→ I totally agree with these comments. I added descriptions in Line80-85 & 289-296.
5) finally, you make a consideration based on histological analysis. Can you state on what basis you reached this conclusion? Was the biopsy performed and evaluated histologically? If so, you should post a picture and comment on it.
→ I totally agree with these comments. This issue is one of the limitations of the present report. I added a description in Line 289-296.
6) the English language needs to be improved.
→ the manuscript has been edited by language editing service. I attached the certificate.
Round 2
Reviewer 1 Report
I have reviewed the re-submission and the authors have carefully amended their manuscript following the additional reviewers' suggestions.
Author Response
We are thankful for the time and energy you expended.
Reviewer 2 Report
There is a 'lost' reference after reference number 25, which the authors need to remove from the manuscript:
"i, E., Akcalı, A., & Donos, N. (2020). The Role of Osteopromotive Membranes in Guided Bone Regeneration. Bone Augmentation by Anatomical Region: Techniques and Decision‐Making, 69-93."
Author Response
We are thankful for the time and energy you expended.
I removed several lines in the reference list.
Thank you very much.
Reviewer 3 Report
Authors were successful in answering the reviewer's comments. Accept in present form.
Author Response

(The authors gave the same response as above.)
